# Differences in the Impact of COVID-19 on Pathology Laboratories and Cancer Diagnosis in Girona

**DOI:** 10.3390/ijerph182413269

**Published:** 2021-12-16

**Authors:** Arantza Sanvisens, Montse Puigdemont, Jordi Rubió-Casadevall, Anna Vidal-Vila, Eugeni López-Bonet, Ferran Martín-Romero, Rafael Marcos-Gragera

**Affiliations:** 1Epidemiology Unit and Girona Cancer Registry, Institut de Recerca Contra la Leucèmia Josep Carreras, 17004 Girona, Spain; asanvisens@carrerasresearch.org; 2Epidemiology Unit and Girona Cancer Registry, Institut Català d’Oncologia, Pla Director d’Oncologia, 17004 Girona, Spain; mpuigdemont@iconcologia.net (M.P.); marianna.vidal@iconcologia.net (A.V.-V.); 3Institut d’Investigació Biomèdica de Girona Dr. Josep Trueta (IDIBGI), 17004 Girona, Spain; jrubio@iconcologia.net; 4Medical Oncology Department, Institut Català d’Oncologia, Hospital Universitari Dr. Josep Trueta, 17007 Girona, Spain; 5School of Medicine, Universitat de Girona, 17004 Girona, Spain; 6Department of Pathology, Hospital Universitari Dr. Josep Trueta, 17007 Girona, Spain; elopezbonet.girona.ics@gencat.cat; 7Department of Pathology, Hospital de Figueres, Fundació Salut Empordà, 17600 Figueres, Spain; lfmartin@salutemporda.cat

**Keywords:** COVID-19, cancer, pathology, diagnosis

## Abstract

Introduction: The recent COVID-19 pandemic has compromised socio-health care, with consequences for the diagnosis and follow-up of other pathologies. The aim of this study was to evaluate the impact of COVID-19 on cancer diagnosis in Girona, Spain. Methodology: Observational study of samples received in two pathology laboratories during 2019–2020 (tertiary hospital in Girona and county hospital in Figueres). Date, sample type, and location and morphology were available. Samples were recoded to determine malignancy and grouped by location. Comparisons were made by calendar year and period of exposure to COVID-19. Results: 102,360 samples were included: 80,517 from Girona and 21,843 from Figueres. The reduction in activity in the pathology laboratories in 2020 compared to the previous year was 25.4% in Girona and 27.5% in Figueres. The reduction in cancer diagnoses in 2020 compared to 2019 was 6.8% in Girona and 21% in Figueres. In both laboratories, a decrease was observed in the diagnoses of neoplasms of the lip, oral cavity and pharynx, larynx, colon, rectum and anus, kidney and urinary system, melanoma, and central nervous system. A statistically significant higher probability of a sample received in the pathology laboratory displaying malignancy during COVID-19 was found (Girona: OR = 1.28, 95% CI: 1.23–1.34; Figueres: OR = 1.10, 95% CI: 1.01–1.20) with respect to the COVID-19-free period. Conclusions: The COVID-19 pandemic has resulted in a reduction in cancer diagnoses by pathology departments that varies according to tumor location and type of hospital. Despite this, the optimization of care resources and the recovery effort have partially reduced the impact of the pandemic in certain neoplasms.

## 1. Introduction

Since the reported appearance of the first cases of SARS-CoV-2 in Spain from March 2020 to July 2021, there have been more than 4 million infections and around 86,000 deaths attributed to the virus [1]. Although the social and economic impact of this disease has been devastating, the health impact is unprecedented in recent decades. Spain was one of the European countries most affected during the first wave of the pandemic [2], but unlike other European countries, no data are yet available to assess the impact the pandemic has had on oncology.

During the first wave of the virus, in order to preserve the care capacity for COVID-19 patients and attempt to control the infection, the Spanish health system was drastically restructured, reducing surgical and care activity dedicated to other diseases. At this point, the emergence of COVID-19 was described as a syndemic, since various diseases were interacting to converge towards a worse health and social situation [3]. The term syndemic stems from the fusion of the terms synergy and epidemic, and was introduced to label the synergistic interaction of two or more coexisting diseases and the resulting excess care and social burden. Currently, this label is easy to apply in the case of cancer, a set of diseases that in themselves are already considered an epidemic and that, to date, has affected 2,265,152 patients in Spain, with an estimated incidence of 277,000 new diagnoses in 2020, bringing with it a heavy care load [4]. In fact, due to COVID-19, screening programs, non-urgent diagnostic procedures, and elective surgery were suspended, and some treatments and visitation programs were also descheduled or modified [5,6].

Generally speaking, cancer prognosis is closely linked to the time of diagnosis and intervention. Given this fact, delays in diagnosis due to COVID-19 may have had a devastating impact, and it is estimated that the excess cancer mortality resulting from this pandemic will reach its maximum peak in the next two years [7]. A study published in January 2021 that analyzed the impact of the pandemic on radiotherapy services in England described an overall reduction of 20%, but with clear differences according to patients’ age and cancer type, reaching a reduction of more than 70% in the case of prostate or non-melanoma skin cancer and, in contrast, an increase in radiotherapy courses for bladder or esophageal cancer [8]. On the other hand, infection with SARS-CoV-2 may also have had a strong impact on the prognosis of cancer patients who are often immunocompromised due to the treatment they receive or the disease itself.

One of the most severely affected cancer prevention and control services has been population-based early detection programs. In Catalonia, the Oncology Master Plan is used to coordinate population cancer screening services. Breast and colorectal cancer screening programs have been temporarily halted to alleviate health care demand due to COVID-19 and little is known about the impact of the current pandemic on cancer detection and prevention activities. The discontinuation of cancer screening programs is expected to lead to an increase in the number of patients with advanced cancer.

The autonomous region of Catalonia has been one of the most compromised by COVID-19 in Spain. The province of Girona, located in the northeast of Catalonia, showed a relatively low prevalence of SARS-CoV-2 in the national study conducted in Spain between April 27 and 11 May 2020, i.e., during the first wave of the epidemic [9]. However, since November 2020, the prevalence of infection in Girona has already exceeded the average for Spain, reaching a prevalence of 8.7% [10]. The aim of this study was to estimate the impact of the pandemic on cancer diagnosis at the population level in Girona by analyzing the activity of two pathology departments in the region.

## 2. Methodology

This study included all specimens processed in the pathology laboratories of two hospitals in the province of Girona between January 2019 and December 2020: (1) Hospital Universitari de Girona Dr. Josep Trueta, which is a third level hospital whose area of influence covers a population comprising the city of Girona and around 156,000 inhabitants, with a population density of 323 inhab/km^2^; it is a tertiary referral hospital for oncology in the province of Girona, and its pathology laboratory also includes the activity generated by Santa Caterina Hospital, and the hospital of Campdevànol-Hospital Comarcal del Ripollès; and (2) Figueres Hospital, which is located within the province of Girona and is a county hospital whose area of influence has a population of 140,000 people from a geographical area with a population density of 104 inhab/km^2^.

The data were extracted completely and irreversibly anonymized from the hospital databases that compile the care activity undertaken by the pathology departments; the dataset was stripped of all identifying information and there is no way that it could be linked back directly or indirectly to the subjects from whom it was originally collected. This study was reviewed by the IRB of the Hospital Universitari de Girona Dr. Josep Trueta (approval number 2021.198). This study did not require informed consent.

In the two participant laboratories, the pathological study included primary samples from autopsies, cytologies, biopsies, and molecular pathology. The results of the pathological study were encoded at Josep Trueta Hospital using SNOMED-CT (Systematized Nomenclature of Medicine—Clinical Terms). SNOMED-CT covers a broad range of health-related topics with comprehensive, scalable, flexible, and internationally controlled vocabulary. The Oncology Master Plan has a pathology subset and microglossary of SNOMED-CT, which has been created using systematically cross-referenced international classifications of diseases, such as the International Classification of Disease Oncology, Third Edition, First revision (ICD-O-3.1), the WHO/IARC classification of tumors series, and the TNM Classification of Malignant Tumors [11,12].

A parallel procedure was used to identify the characteristics of the samples (topography, morphology, and behavior when a tumor was present) from Figueres Hospital, the initial encoding of which was performed using SNOMED II, that is a previous version of SNOMED-CT.

Using the pathology subset and microglossary from the Oncology Master Plan, the SNOMED-CT and SNOMED II codes were recoded to identify cancer patients and obtain topography and morphology according to ICD-O-3.1 [13]. In addition to the topographic and morphological description, when tumors were present, it was determined whether these were in situ, infiltrating, or metastatic.

The pathology department at the Hospital Universitari de Girona Dr. Josep Trueta recorded a total of 118,653 specimens during the study period, 65,132 in 2019 and 53,521 in 2020. The median number of samples per specimen was 1 [interquartile range (IQR): 1–2], range 1–39 samples. At Figueres Hospital, 41,526 specimens were recorded (23,529 in 2019 and 17,997 in 2020), with a median number of samples per specimen of 1 [IQR: 1–2] and range 1–20. For the purposes of this study, samples with duplicate results and those samples with metastatic results were excluded if primary tumor information was available in the same specimen. In the case of samples of the same specimen with discordant results, the one with the worst known code of conduct according to ICD-O-3.1 was selected [13].

Neoplasms (malignant behaviors according to ICD-O-3.1) were classified using the topographic and morphological code and grouped as follows: (1) Hematopoietic and reticuloendothelial systems and lymph nodes (morphological code 959–999); (2) lip, oral cavity, and pharynx (C00–C14); (3) esophagus (C15); (4) stomach (C16); (5) colon, rectosigmoid juction, rectum, anus, and anal canal (C18–C21); (6) liver and intrahepatic bile ducts (C22); (7) biliary tract and gallbladder (C23–C24); (8) pancreas (C25); (9) larynx (C32); (10) trachea, bronchus, and lung (C33–C34); (11) bones, joints, and articular cartilage (C40–C41); (12) skin melanoma (C44 with morphological codes 8720–8790); (13) breast C50; (14) cervix uteri C53; (15) corpus uteri (C54); (16) ovary (C56); (17) prostate gland (C61); (18) testis (C62); (19) kidney, renal pelvis, ureter, other unspecified urinary organs (C64–C66, C68); (20) bladder (C67); (21) central nervous system (C70–C72); (22) thyroid gland (C73); (23) unknown primary site (C80); (24) all codes not provided in previous classifications.

Samples that were not malignant were classified according to the topographic code following the same categories.

Data referring to COVID-19 hospitalizations in the geographical areas of reference for the hospitals participating in this study were obtained from the Autonomous Government of Catalonia’s official open data portal, using the set of daily COVID-19 data by county [14]. Monitoring of COVID-19 hospitalizations in the geographic area under study was available from 20 April 2020 onwards (week 17).

### Statistical Analysis

A descriptive analysis was performed of the activity carried out by the pathology laboratories in 2019 and 2020, obtaining both overall and weekly figures. Descriptive statistics were expressed as the median (IQR) for quantitative variables and as absolute frequencies and percentages for qualitative variables. The percentage variation in neoplasia diagnoses from 2019 to 2020 was calculated overall and according to topographic sites due to the variability between types of cancer in terms of cancer detection (i.e., screening programs, incidentals, symptomatology) and the use of diagnostic methods without histological confirmation (i.e., specific tumor markers or clinical findings).

In order to analyze the specific impact of COVID-19 on each cancer type, two periods were determined: the COVID-19-free period (January 2019 to February 2020) and the COVID-19 exposure period (March to December 2020).

The probability (OR and 95% confidence interval (CI)) of observing a diagnosis of neoplasia in samples collected during the period of COVID-19 exposure was calculated with respect to the non-COVID-19 period for each sample location group.

*p* values < 0.05 were considered statistically significant. Statistical analyses were performed using Stata software (version 11.1, StataCorp LLC, College Station, TX, USA).

## 3. Results

A total of 102,360 samples were included in the study, 80,517 from the Dr. Josep Trueta Hospital in Girona and 21,843 from Figueres Hospital. With regard to the source of the samples, 46% of the included samples came from biopsies and 43% from cytologies. Table 1 shows the distribution of samples by year, type of analysis, and center. Overall, the reduction in activity in the pathology laboratories in 2020 compared to the previous year was 25% in Girona and 27% in Figueres. As Figure 1a shows, during the months of March and April 2020 (weeks 12–22), which coincided with the first wave of COVID-19 in Spain and the first lockdown, the reduction in activity in the services was as high as 66% in Girona and 75% in Figueres.

In the Girona hospital, 9004 cases of neoplasms were identified, 4661 in 2019 and 4343 in 2020, representing a 6.8% decrease over that period. In the lockdown period, the reduction in neoplasia diagnoses was 39%. Figure 1b shows the weekly distribution of cancer diagnoses in 2019 and 2020. The variation in neoplasm diagnoses from 2020 to 2019 differed greatly depending on the type of tumor. While an increase was observed in tumors of the gallbladder and bile ducts (29%), urinary bladder (13%), pancreas (17%), bone (83%), and ovaries (61%), a reduction was observed in tumors of the testicles (100%), thyroid (40%), prostate (36%), stomach (24%), larynx (21%), and central nervous system (22%), among others. Table 2 shows the distribution of neoplastic diagnoses by type and year of diagnosis.

In the case of Figueres Hospital, 2347 cases of neoplasms were identified, 1309 in 2019 and 1038 in 2020, representing a 21% decrease in 2020 compared to 2019. In the lockdown period, the reduction in diagnoses of neoplasia was 39% (Figure 1b). Regarding the location of tumors, there was an increase in tumors of the stomach (67%); cervix (25%); trachea, bronchi, and lung (21%); liver (11%); hematological (6.2%); and those with unknown location (46%). For all other locations, a reduction in diagnoses was observed in 2020 compared to 2019 (Table 2).

Analyzing the impact of COVID-19 from the moment the first cases of the infection were officially detected in Spain (March 2020) with respect to the COVID-19-free period (between January 2019 and February 2020), the probability that a sample received in the pathology laboratory would display malignancy during the COVID-19 period compared to the COVID-19-free period was higher for both hospitals (Girona: OR = 1.28, 95% CI: 1.23–1.34, *p* < 0.001; Figueres: OR = 1.10, 95% CI: 1.01–1.20, *p* = 0.02).

Figure 2 shows the probability of diagnosing cancer according to the location/morphology of the sample during the COVID-19 period with respect to the COVID-19-free period. Specifically, in Girona, the probability of diagnosing cancer in a sample of lip, oral cavity, and pharynx (OR = 1.4, 95% CI: 1.02–1.94, *p* = 0.04); colon, rectum, and anus (OR = 1.22, 95% CI: 1.06–1.04, *p* = 0.005); trachea, bronchi, and lung (OR = 1.33, 95% CI: 1.12–1.59, *p* = 0.001); breast (OR = 1.31, 95% CI: 1.11–1.54, *p* = 0.001); ovary (OR = 2.46, 95% CI: 1.42–4.24, *p* = 0.001); or skin melanoma (OR = 1.80, 95% CI: 1.13–2.85, *p* = 0.01) was higher during the period with COVID-19. In contrast, a lower probability of diagnosis was observed in the hospital in Figueres in the case of colon, rectum, and anus cancer during the COVID-19 period compared to the COVID-19-free period (OR = 0.55, 95% CI: 0.39–0.79, *p* = 0.001).

## 4. Discussion

The results of this study show the impact of the COVID-19 pandemic on cancer diagnosis in a tertiary referral hospital for oncology and in another local hospital in the province of Girona. A decrease in diagnoses of neoplasia was found in 2020 compared to 2019; however, clear differences are observed depending on hospital type and tumor type.

In the case of the tertiary referral hospital in Girona, which is a reference center for oncology in the province, a reduction in cancer diagnoses of around 6% was detected at the end of 2020, and in Figueres Hospital, which is a county center, there was a 20% reduction in cancer diagnoses from 2019 to 2020 overall. To the best of our knowledge, the only study to have analyzed cancer data based on pathology results in Spain observed a 17% reduction in diagnoses, this finding being similar to those of other studies [15,16]. These figures reflect the effort to recover care services, especially oncology referrals, despite their saturation after reaching a 70% reduction in activity during the first wave, as other studies have shown [17,18,19]. In addition, despite there being less activity in the pathology departments during COVID-19 than during 2019, the probability of neoplasia diagnosis during the pandemic has been statistically significantly higher. Although this could be due to changes in the prioritization protocols used by different care services and recommendations by scientific societies and expert groups to minimize the risks [20,21,22], in certain types of cancer, it could also be due to the increase in the number of scans due to COVID-19 in patients with undiagnosed cancers. In fact, patients with underlying cancers are probably more vulnerable to SARS-CoV-2 infection and display a worse progression of the disease [23,24,25]. Therefore, it is plausible that people with cancer have a higher probability of having required hospitalization during the COVID-19 pandemic than others. In this sense, there has been a greater probability of diagnosing certain neoplasms in the tertiary referral hospital in Girona during the pandemic, such as those located in the lip, oral cavity, and pharynx; trachea, bronchi, and lungs; breast; ovary; or melanoma, with most of these requiring an exploration of the respiratory system while testing for COVID-19. In the case of melanoma, the higher probability could be due to an increase of dermatology exploration related to cutaneous manifestations in the context of COVID-19, which have been reported in nearly 20% of COVID-19 hospitalized patients [26,27].

A comparison of the number of neoplasia diagnoses between 2019 and 2020 revealed marked differences between the participating centers, probably due to the type of care resources at the oncological level and the low frequency of certain tumors. However, the results discussed below were similar for both centers. Specifically, a reduction was observed in the pathology diagnoses of neoplasms of the lip, oral cavity, and pharynx; larynx, colon, rectum, and anus; kidney, melanoma, prostate, and urinary system; as well as the central nervous system. With particular regard to tumors of the central nervous system and the head and neck, it is possible that much of the diagnoses were based solely on imaging tests and that more conservative treatments with radiation therapy were opted for given the reduction in the number of surgeries, and the difficulty of accessing high-tech operating rooms and consequently in obtaining tumor samples in certain locations during COVID-19 [28,29]. Nevertheless, discrepant data have been found on the use of radiation therapy during the pandemic to date. One study evaluating the impact of COVID-19 on radiotherapy services in the United Kingdom between February and June 2020 compared to the same time period from the previous year did not observe an increase in activity in relation to neoplasms in these locations [8], contrary to that observed by He et al. for head and neck tumors in China [30]. This group of tumors has shown a relatively low five-year survival rate of around 50% or less, a figure that could worsen in the coming years due to diagnostic and/or therapeutic delays.

The decrease in colon, rectal, and anus cancer may be linked to the suspension of the screening program in the province of Girona between March and August 2020. In fact, several studies have quantified the reduction of screenings, both breast and colorectal, at between 40% and 80% from 2019 to 2020, depending on the time of interruption [17,31,32,33,34,35]. In this regard, it is surprising not to see a reduction in the diagnosis of breast cancer, although the loss of screening tests could be offset by the increase in chest examinations due to COVID-19, as some studies have noted [16,30,36].

As for kidney and prostate cancer, the reduction may be due to the fact that it is a type of neoplasm that is mostly asymptomatic and usually diagnosed incidentally. People with nonspecific symptoms of cancer have had difficulty accessing the primary care physician’s office during the pandemic. Particularly in the early months, most primary care consultations turned to telemedicine, and hospitals postponed complementary diagnostic evaluation tests by focusing their resources on COVID-19. The above, as well as the fear of possible infections when attending health centers and the moral concern to not oversaturate already overwhelmed care services, may have also influenced this decrease in diagnoses. The reduction in diagnoses of these types of cancer could result in an increase in patients diagnosed with advanced cancer in the coming years and, therefore, an increase in more aggressive therapeutic strategies and/or worse survival.

A reduction in prostate and thyroid diagnoses was also observed for 2020 compared to 2019. These results make sense and are consistent with others that have been published [37], such as those related to melanoma, given that these types of neoplasms do not require urgent surgical intervention and generally have a good prognosis. In the case of prostate cancer, the reduction in diagnoses may be in part due to the decline in prostate-specific antigen (PSA) testing observed in recent years [38]. Whatever the cause, it should be borne in mind that a delay in surgery may favor the progression of a tumor to the point where it becomes incurable [21,39].

In contrast with the above, there was an increase in the diagnosis of cancer of the cervix, corpus uteri, and liver in both services in 2020 compared to 2019. With regard to the liver, the increase in diagnoses is consistent with the increasing trend observed in this type of neoplasm [40,41], indicating that the diagnosis of this type of cancer has not been affected by the context in our health area. This contrasts with the results of a study analyzing data from 76 international centers in relation to care for liver cancer patients, which confirmed that 87% of services modified their clinical practice protocols and 40.8% specifically modified their diagnostic procedure system [42]. Finally, the interpretation of the increase in cervical and uterine tumors is limited since there was a change in the gynecological care protocols in Girona due to the creation of a specific oncological gynecology unit, probably leading to an absence of this cancer diagnosis in county hospitals. This study has other limitations that should also be mentioned. Firstly, some of the biopsies encoded as nonspecific could correspond to patients with a known primary tumor prior to the study period, especially those from the first quarter of 2019. On the other hand, no sociodemographic variables are available (age, sex, socioeconomic level, among others), or information on other methods of cancer diagnosis, which could have helped clarify some findings of the study [18,43,44,45,46]. Finally, the different nature of the two hospitals participating in this study does not allow for a direct comparison of the results obtained in separate pathology laboratories, since one of them is a reference for oncology in the province. However, these same differences in origin do allow us to measure the impact of COVID-19 in two types of hospitals with marked differences in oncology care.

## 5. Conclusions

In conclusion, the COVID-19 pandemic has resulted in a reduction in cancer diagnoses by pathology laboratories that varies according to tumor location and type of hospital. The results of this study suggest a diagnostic delay in certain types of cancer, such as colorectal or urinary tract tumors, as well as a delay in elective surgery on critical tumors, such as those of the central nervous system or head and neck, which may worsen prognosis. However, it is possible that the use of specific tumor markers or other diagnostic techniques including X-ray, imaging, or ultrasound without waiting for histological confirmation, as well as new therapeutic protocols adapted to the new health situation [21,47], may have partially reduced the impact of the pandemic in certain neoplasms. In fact, the data in this study show the great efforts made to get the health system back on its feet and optimize resources, given the increase in the number of biopsied malignant samples during this unprecedented time. Finally, it is worth noting that population studies will be needed to determine the real impact of the epidemic on the diagnosis and incidence of cancer.

## Figures and Tables

**Figure 1 ijerph-18-13269-f001:**
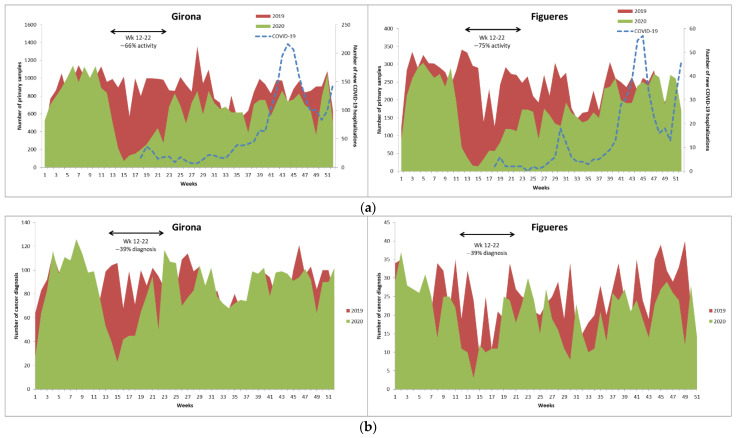
Weekly distribution of (**a**) activity of pathology departments and (**b**) cancer diagnoses in 2019 and 2020.

**Figure 2 ijerph-18-13269-f002:**
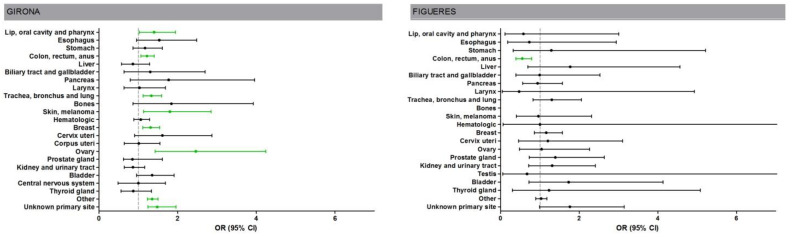
Probability of diagnosing cancer according to the location/morphology of the sample during the COVID-19 period with respect to the COVID-19-free period.

**Table 1 ijerph-18-13269-t001:** Description of activity in two pathology laboratories in Girona, 2019–2020.

	Girona		Figueres	
	2019 N = 46,107	2020 N = 34,410	2019 N = 12,664	2020 N = 9179
Type of analysis				
Autopsy	58 (0.1)	51 (0.1)	17 (0.1)	21 (0.2)
Biopsy	20,178 (44)	16,212 (47)	6243 (49)	4740 (52)
Cytology	19,981 (43)	13,336 (39)	6285 (50)	4289 (47)
Molecular pathology/Immunohistochemistry	5890 (13)	4811 (14)	119 (0.9)	129 (1.4)

**Table 2 ijerph-18-13269-t002:** Distribution of neoplastic diagnoses by type and year of diagnosis.

	Girona					Figueres				
	2019		2020		Differential 2020 vs. 2019	2019		2020		Differential 2020 vs. 2019
	N	%	N	%	%	N	%	N	%	%
Lip, oral cavity, and pharynx (C00–C14)	95	2.0	85	1.9	−10	6	0.5	3	0.3	−50
Esophagus (C15)	39	0.8	39	0.9	0	9	0.7	4	0.4	−56
Stomach (C16)	101	2.2	77	1.8	−24	3	0.2	5	0.5	67
Colon, rectum, anus (C18–C21)	530	11	443	10	−16	81	6.2	65	6.3	−20
Liver (C22)	85	1.8	89	2.0	4.7	9	0.7	10	1.0	11
Biliary tract and gallbladder (C23–C24)	14	0.3	18	0.4	29	13	1.0	7	0.7	−46
Pancreas (C25)	30	0.6	35	0.8	17	42	3.2	30	2.9	−29
Larynx (C32)	61	1.3	48	1.1	−21	4	0.3	1	0.1	−75
Trachea, bronchus, and lung (C33–C34)	443	9.5	412	9.5	−7.0	99	7.6	120	12	21
Bones (C40–C41)	12	0.3	22	0.5	83	4	0.3	1	0.1	−75
Skin, melanoma	44	0.9	39	0.9	−11	16	1.2	10	1.0	−37
Hematologic (C42, C77)	391	8.4	367	8.4	−6.1	32	2.4	34	3.3	6.2
Breast (C50)	494	11	528	12	6.9	179	14	154	15	−14
Cervix uteri (C53)	23	0.5	24	0.5	4.3	8	0.6	10	1.0	25
Corpus uteri (C54)	37	0.8	44	1.0	19	0	0	0	0	-
Ovary (C56)	28	0.6	45	1.0	61	18	1.4	12	1.2	−33
Prostate gland (C61)	259	5.6	165	3.8	−36	36	2.7	34	3.3	−5.6
Testis (C62)	5	0.1	0	0	−100	2	0.1	2	0.2	0
Kidney and urinary tract (C64–C66, C68)	106	2.3	90	2.1	−15	26	2.0	22	2.1	−15
Bladder (C67)	136	2.9	154	3.5	13	76	5.8	73	7.0	−3.9
Central nervous system (C70–C72)	86	1.8	67	1.5	−22	0	0	0	0	-
Thyroid gland (C73)	65	1.4	39	0.9	−40	6	0.5	3	0.3	−50
Other (rest of codes)	1149	25	1127	26	−1.9	612	47	397	38	−36
Unknown primary site (C80)	428	9.2	386	8.9	−9.8	28	2.1	41	3.9	46
Total	4661	100	4343	100	−6.8	1309	100	1038	100	−21

## Data Availability

The dataset analyzed during the current study is not publicly available due to national regulations of cancer registry data. However, it is available anonymized from Rafael Marcos-Gragera (rmarcos@iconcologia.net) on reasonable request.

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
