# Peer review of "Differences in the Impact of COVID-19 on Pathology Laboratories and Cancer Diagnosis in Girona"

_ijerph, 2021, doi:10.3390/ijerph182413269_

Round 1

Reviewer 1 Report

The manuscript reports an observational study of samples received by two pathology laboratories in the province of Girona, Spain, during the period from 2019 to 2020, revealing a significant reduction in the activity of both laboratories in the latter compared to the former year besides a statistically significant higher probability of a sample displaying malignancy during the COVID-19 epidemic. Although date, sample type and location/morphology data were shown in the results section, sociodemographic variables and information on other methods of cancer diagnosis were unfortunately unavailable and indicated as a limitation of the study by the authors themselves in the discussion section. Furthermore, the manuscript lacks statements regarding the approval of its research protocol by the institutional review board of both hospitals where the pathology laboratories are located and the signing of informed consent forms by the participants.

Author Response

Reviewer 1

Comments and Suggestions for Authors

The manuscript reports an observational study of samples received by two pathology laboratories in the province of Girona, Spain, during the period from 2019 to 2020, revealing a significant reduction in the activity of both laboratories in the latter compared to the former year besides a statistically significant higher probability of a sample displaying malignancy during the COVID-19 epidemic. Although date, sample type and location/morphology data were shown in the results section, sociodemographic variables and information on other methods of cancer diagnosis were unfortunately unavailable and indicated as a limitation of the study by the authors themselves in the discussion section. Furthermore, the manuscript lacks statements regarding the approval of its research protocol by the institutional review board of both hospitals where the pathology laboratories are located and the signing of informed consent forms by the participants.

Thank you for the review. Indeed, as it is totally anonymized data that comes from the activity of pathological anatomy laboratories, there is no specific information on the individuals, only the information on the specimens was obtained. In this sense, informed consent was not obtained and the study did not require approval by the institutional review board. However, this clarification has been added to the paper as follows:

Methodology section, page 3, second paragraph:

“…The data were extracted completely and irreversibly anonymized from the hospital databases that compile the care activity undertaken by the pathology departments; the dataset were stripped of all identifying information and there is no way that it could be linked back directly or indirectly to the subjects from whom it was originally collected. Therefore, the study neither required informed consent nor institutional review board…”

Reviewer 2 Report

This manuscript explores the changes in cancer diagnosis and pathology of samples at laboratories in two cities in Spain in the COVID-19 era. A lower rate of cancer diagnosis was found along with a higher rate of samples sent to pathology departments having cancer. Rates varied by location and by cancer type.

Affiliation: Please include the region and country of Girona. I had no idea where it was.

Girona is a city in northern Catalonia, Spain,

A statistically significant higher probability of a sample received in 32
the pathology laboratory displaying malignancy during COVID-19 was found (Girona: OR=1.28, 33
95%CI: 1.23-1.34; Figueres: OR=1.10, 95%CI: 1.01-1.20) with respect to the COVID-19 free period. 34
Conclusions: The COVID-19 pandemic has resulted in a reduction in cancer diagnoses by pathol- 35
ogy departments that varies according to tumour location and type of hospital. Despite this, the 36
optimization of care resources and recovery effort have partially mitigated the damage caused.

Comment: Is it possible that some of the increase in sample malignancy was due to people with cancer dying from COVID? Both most cancers and COVID-19 are strongly associated with low serum 25(OH)D concentration based on ecological, observational, and supplementation studies as well as an understanding of the mechanisms. Suggest discussing this possibility in the manuscript.

In this sense,
235 there has been a greater probability of diagnosing certain neoplasms in the
236 tertiary referral hospital in Girona during the pandemic, such as those locat-
237 ed in the lip, oral cavity and pharynx, trachea, bronchi and lungs, breast,
238 ovary or melanoma, most of these requiring an exploration of the respiratory
239 system while testing for COVID-19\

Comment: Does not explain melanoma. Perhaps since melanoma is a surface cancer, people were more likely to stay home and have time to examine their bodies.

Additional publications to consider citing found through a quick search at scholar.google.com

Perhaps some of the data could be put in tabular form. Additional related publications could also be included.

The impact of the COVID-19 pandemic on cancer deaths due to delays in diagnosis in England, UK: a national, population-based, modelling study.

Maringe C, Spicer J, Morris M, Purushotham A, Nolte E, Sullivan R, Rachet B, Aggarwal A.Lancet Oncol. 2020 Aug;21(8):1023-1034. doi: 10.1016/S1470-2045(20)30388-0.

Impact of the COVID-19 Pandemic on Breast Cancer Mortality in the US: Estimates From Collaborative Simulation Modeling.

Alagoz O, Lowry KP, Kurian AW, Mandelblatt JS, Ergun MA, Huang H, Lee SJ, Schechter CB, Tosteson ANA, Miglioretti DL, Trentham-Dietz A, Nyante SJ, Kerlikowske K, Sprague BL, Stout NK.J Natl Cancer Inst. 2021 Jul 14:djab097. doi: 10.1093/jnci/djab097.

Disparities in Cancer Prevention in the COVID-19 Era.

Carethers JM, Sengupta R, Blakey R, Ribas A, D'Souza G.Cancer Prev Res (Phila). 2020 Nov;13(11):893-896. doi: 10.1158/1940-6207.CAPR-20-0447.

An inverse stage-shift model to estimate the excess mortality and health economic impact of delayed access to cancer services due to the COVID-19 pandemic.

Degeling K, Baxter NN, Emery J, Jenkins MA, Franchini F, Gibbs P, Mann GB, McArthur G, Solomon BJ, IJzerman MJ.Asia Pac J Clin Oncol. 2021 Aug;17(4):359-367. doi: 10.1111/ajco.13505.

A clinical dilemma amid COVID-19 pandemic: missed or encountered diagnosis of cancer?

Yekedüz E, Karcıoğlu AM, Utkan G, Ürün Y.Future Oncol. 2020 Sep;16(25):1879-1881. doi: 10.2217/fon-2020-0501. 

Impact of COVID-19 on Cervical Cancer Screening Rates Among Women Aged 21-65 Years in a Large Integrated Health Care System - Southern California, January 1-September 30, 2019, and January 1-September 30, 2020.

Miller MJ, Xu L, Qin J, Hahn EE, Ngo-Metzger Q, Mittman B, Tewari D, Hodeib M, Wride P, Saraiya M, Chao CR.MMWR Morb Mortal Wkly Rep. 2021 Jan 29;70(4):109-113. doi: 10.15585/mmwr.mm7004a1.

Impact of the COVID-19 pandemic during Spain's state of emergency on the diagnosis of colorectal cancer.

Suárez J, Mata E, Guerra A, Jiménez G, Montes M, Arias F, Ciga MA, Ursúa E, Ederra M, Arín B, Laiglesia M, Sanz A, Vera R.J Surg Oncol. 2021 Jan;123(1):32-36. doi: 10.1002/jso.26263. 

Impact of COVID-19 on Cancer Care: How the Pandemic Is Delaying Cancer Diagnosis and Treatment for American Seniors.

Patt D, Gordan L, Diaz M, Okon T, Grady L, Harmison M, Markward N, Sullivan M, Peng J, Zhou A.JCO Clin Cancer Inform. 2020 Nov;4:1059-1071. doi: 10.1200/CCI.20.00134.

Significant digits. The general rule is that no more non-zero digits should be given than are justified by the uncertainty of the value.

See "Too many digits: the presentation of numerical data"

https://www.ncbi.nlm.nih.gov/pmc/articles/PMC4483789/

If the uncertainty is greater than about 7%, only two non-zero digits are justified.

P values should be given to two decimal places unless the first two are 00 or the number lies between 0.045 and 0.054.

Thus, in Table 2, most percentages should be given in whole numbers since N is generally less than 150. Percentages for N >150 could be given with one or, sometimes, two decimal places, but that is extra information than is not needed. Readers could calculate the decimal places if desired.

Please review all numbers in abstract, text, tables, and figures and adjust accordingly

Author Response

Reviewer 2

This manuscript explores the changes in cancer diagnosis and pathology of samples at laboratories in two cities in Spain in the COVID-19 era. A lower rate of cancer diagnosis was found along with a higher rate of samples sent to pathology departments having cancer. Rates varied by location and by cancer type.

Affiliation: Please include the region and country of Girona. I had no idea where it was.

Girona is a city in northern Catalonia, Spain,

We have included the country in the affiliations. 

A statistically significant higher probability of a sample received in the pathology laboratory displaying malignancy during COVID-19 was found (Girona: OR=1.28,
95%CI: 1.23-1.34; Figueres: OR=1.10, 95%CI: 1.01-1.20) with respect to the COVID-19 free period.
Conclusions: The COVID-19 pandemic has resulted in a reduction in cancer diagnoses by pathology departments that varies according to tumour location and type of hospital. Despite this, the  optimization of care resources and recovery effort have partially mitigated the damage caused.

Comment: Is it possible that some of the increase in sample malignancy was due to people with cancer dying from COVID? Both most cancers and COVID-19 are strongly associated with low serum 25(OH)D concentration based on ecological, observational, and supplementation studies as well as an understanding of the mechanisms. Suggest discussing this possibility in the manuscript.

The reviewer's comment is interesting and we agree that it could be that a part of the new cancer diagnoses in 2020 corresponded to patients dying from COVID-19 in the study hospitals; unfortunately, we do not have mortality data on patients and the number of autopsies is similar in 2019/2020; In this sense, we believe it is more cautious to speak in terms of "higher need of medical care" than "mortality."

Therefore, we have modified the Discussion section (page 9, second paragraph) as follows:

“...In fact, patients with underlying cancers are probably more vulnerable to SARS-COV-2 infection and display a worse progression of the disease 23–25. Therefore, it is plausible that people with cancer have higher probability to have been required hospitalization during COVID-19 pandemic than others...”

Related to 25(OH)D, there are an interesting discussion regarding the literature on this topic. However, there is a controversy in some results related to Vitamin D and cancer prognosis (Gnagnarella P et al. Nutrients 2021). On the other hand, we believe that it would not be appropriate to discuss this topic in our article since no results on laboratory parameters are displayed.

In this sense, there has been a greater probability of diagnosing certain neoplasms in the tertiary referral hospital in Girona during the pandemic, such as those located in the lip, oral cavity and pharynx, trachea, bronchi and lungs, breast, ovary or melanoma, most of these requiring an exploration of the respiratory system while testing for COVID-19\

Comment: Does not explain melanoma. Perhaps since melanoma is a surface cancer, people were more likely to stay home and have time to examine their bodies.

I think this point has been misunderstood. As can be seen in Table 2, globally there is a reduction in melanoma diagnoses in 2020 compared to 2019 in both hospitals and this would be consistent with the reviewer's comment. However, the probability that a skin sample that arrived at the laboratory was malignant was higher during the COVID-19 era.

To clarify this aspect we have expanded the related paragraph in the Discussion section (page 9) as follows:

“…In this sense, there has been a greater probability of diagnosing certain neoplasms in the tertiary referral hospital in Girona during the pandemic, such as those located in the lip, oral cavity and pharynx, trachea, bronchi and lungs, breast, ovary or melanoma, most of these requiring an exploration of the respiratory system while testing for COVID-19. In the case of melanoma, the higher probability could be due to an increase of dermatology exploration related to cutaneous manifestations in the context of COVID-19, that have been reported in near 20% of COVID-19 hospitalized patients 26,27.…”

Additional publications to consider citing found through a quick search at scholar.google.com Perhaps some of the data could be put in tabular form. Additional related publications could also be included.

The impact of the COVID-19 pandemic on cancer deaths due to delays in diagnosis in England, UK: a national, population-based, modelling study.

Maringe C, Spicer J, Morris M, Purushotham A, Nolte E, Sullivan R, Rachet B, Aggarwal A.Lancet Oncol. 2020 Aug;21(8):1023-1034. doi: 10.1016/S1470-2045(20)30388-0.

Impact of the COVID-19 Pandemic on Breast Cancer Mortality in the US: Estimates From Collaborative Simulation Modeling.

Alagoz O, Lowry KP, Kurian AW, Mandelblatt JS, Ergun MA, Huang H, Lee SJ, Schechter CB, Tosteson ANA, Miglioretti DL, Trentham-Dietz A, Nyante SJ, Kerlikowske K, Sprague BL, Stout NK.J Natl Cancer Inst. 2021 Jul 14:djab097. doi: 10.1093/jnci/djab097.

Disparities in Cancer Prevention in the COVID-19 Era.

Carethers JM, Sengupta R, Blakey R, Ribas A, D'Souza G.Cancer Prev Res (Phila). 2020 Nov;13(11):893-896. doi: 10.1158/1940-6207.CAPR-20-0447.

An inverse stage-shift model to estimate the excess mortality and health economic impact of delayed access to cancer services due to the COVID-19 pandemic.

Degeling K, Baxter NN, Emery J, Jenkins MA, Franchini F, Gibbs P, Mann GB, McArthur G, Solomon BJ, IJzerman MJ.Asia Pac J Clin Oncol. 2021 Aug;17(4):359-367. doi: 10.1111/ajco.13505.

A clinical dilemma amid COVID-19 pandemic: missed or encountered diagnosis of cancer?

Yekedüz E, Karcıoğlu AM, Utkan G, Ürün Y.Future Oncol. 2020 Sep;16(25):1879-1881. doi: 10.2217/fon-2020-0501. 

Impact of COVID-19 on Cervical Cancer Screening Rates Among Women Aged 21-65 Years in a Large Integrated Health Care System - Southern California, January 1-September 30, 2019, and January 1-September 30, 2020.

Miller MJ, Xu L, Qin J, Hahn EE, Ngo-Metzger Q, Mittman B, Tewari D, Hodeib M, Wride P, Saraiya M, Chao CR.MMWR Morb Mortal Wkly Rep. 2021 Jan 29;70(4):109-113. doi: 10.15585/mmwr.mm7004a1.

Impact of the COVID-19 pandemic during Spain's state of emergency on the diagnosis of colorectal cancer.

Suárez J, Mata E, Guerra A, Jiménez G, Montes M, Arias F, Ciga MA, Ursúa E, Ederra M, Arín B, Laiglesia M, Sanz A, Vera R.J Surg Oncol. 2021 Jan;123(1):32-36. doi: 10.1002/jso.26263. 

Impact of COVID-19 on Cancer Care: How the Pandemic Is Delaying Cancer Diagnosis and Treatment for American Seniors.

Patt D, Gordan L, Diaz M, Okon T, Grady L, Harmison M, Markward N, Sullivan M, Peng J, Zhou A.JCO Clin Cancer Inform. 2020 Nov;4:1059-1071. doi: 10.1200/CCI.20.00134.

We thank the reviewer for the list of articles. While the scientific literature on COVID-19 is being very extensive we have added some of these bibliographic citations in the new version of the article such as Suarez et al. J Surg Oncol 2021, Yekedüz E et al. Future Oncol 2020.

Significant digits. The general rule is that no more non-zero digits should be given than are justified by the uncertainty of the value.

See "Too many digits: the presentation of numerical data"

https://www.ncbi.nlm.nih.gov/pmc/articles/PMC4483789/

If the uncertainty is greater than about 7%, only two non-zero digits are justified.

P values should be given to two decimal places unless the first two are 00 or the number lies between 0.045 and 0.054.

Thus, in Table 2, most percentages should be given in whole numbers since N is generally less than 150. Percentages for N >150 could be given with one or, sometimes, two decimal places, but that is extra information than is not needed. Readers could calculate the decimal places if desired.

Please review all numbers in abstract, text, tables, and figures and adjust accordingly

We have reviewed all numbers throughout the document following the reviewer suggestion. Please note, that we attach a new version of Figure 1 that must be replaced.

Reviewer 3 Report

As a result of the massive outbreak of COVID-19 in Spain, the local public health organizations took a series of measures including suspending screening programmes, non-urgent diagnostic procedures and elective surgery, and some treatments and visitation programmes were also descheduled or modified. The authors obtained relevant cancer diagnosis data from two local hospitals and they found that delays in tumor diagnosis caused by above taken measures could  have a serious impact on patients and increase the probability of samples malignancy. In summary, the authors systematically assess the impact of the COVID-19 pandemic on cancer diagnosis, but the results of the authors' analysis do not seem to prove their points, as summarized below:

  1. The authors found a 25.4% and5% decrease comparingto the previous year in pathology laboratory activity in Girona and Figueres, respectively. Then, they analyzed that the variation in neoplasm diagnoses from 2020 to 2019 differed greatly depending on the type of tumor. If the delay in diagnosis is due to suspension of screening and treatment activities, why do the delays depend on the type of tumor? The author did not explain reasons in the subsequent analysis. If you can analyze the reasons leading to this phenomenon, it will be more convincing.

  1. The authors point out that the COVID-19 pandemic has increasedthe probability of tumors malignancy without any presented statistical data. If you can give relevant statistical data, it will be more credible.

  1. Although the authors may makelots of efforts in counting the tumor samples, they didn’t do enough work except for presenting statistical data they obtained and analyzing the impact of the COVID-19 pandemic on cancer diagnosis. It is obviously that the delayed diagnosis may increase the probability of tumors malignancy, you should give more points by the further analysis of samples and explain the reasons. In addition, the impact of COVID-19 on cancer patients should be discussed.

  1. The impact of COVID-19 on patients goes far beyond delayed diagnosis, the SARS-CoV-2 infection mayalso affect the proliferation of tumor cells by the activation of further immune reactions or the further infection may make patients’ condition worse, these are issues of more concerns. You should take advantage of the obtained resources and get more conclusions.

In summary, this article still needs to be supplemented and improved, you should further explain the reasons to make your views convinced and make a in-depth analysis of samples to have more precious conclusions. Hope to see your rewritten manuscripts.

Author Response

Reviewer 3

As a result of the massive outbreak of COVID-19 in Spain, the local public health organizations took a series of measures including suspending screening programmes, non-urgent diagnostic procedures and elective surgery, and some treatments and visitation programmes were also descheduled or modified. The authors obtained relevant cancer diagnosis data from two local hospitals and they found that delays in tumor diagnosis caused by above taken measures could  have a serious impact on patients and increase the probability of samples malignancy. In summary, the authors systematically assess the impact of the COVID-19 pandemic on cancer diagnosis, but the results of the authors' analysis do not seem to prove their points, as summarized below:

 1. The authors found a 25.4% and 5% decrease comparing to the previous year in pathology laboratory activity in Girona and Figueres, respectively. Then, they analyzed that the variation in neoplasm diagnoses from 2020 to 2019 differed greatly depending on the type of tumor. If the delay in diagnosis is due to suspension of screening and treatment activities, why do the delays depend on the type of tumor? The author did not explain reasons in the subsequent analysis. If you can analyze the reasons leading to this phenomenon, it will be more convincing.

The diagnostic delay of cancer during the temporary suspension of part of the care activity may depend on the type of tumor for various reasons. Population screening programs only exist for colon and breast cancer in our geographical area. Some tumors are silent in their debut and others the opposite; This means that, during the time of COVID-19 with a reduction of healthcare activity and other restrictive measures, access to the health system may have been unequal depending on the symptoms of the patient who debuts with cancer. It is for this reason that an analysis has been made according to the type of tumor and an attempt is made to give possible explanations in the discussion. To clarify why this type of analysis was carried out, the following sentence has been added in the statistical analysis section (page 4, first paragraph):

“…The percentage variation in neoplasia diagnoses from 2019 to 2020 was calculated overall and according topographic sites due to the variability between types of cancer in terms of cancer detection (i.e. screening programmesprograms, incidentals, symptomatology) and the use of diagnostic methods without histological confirmation (i.e. specific tumour markers or clinical findings)….”

 2. The authors point out that the COVID-19 pandemic has increased the probability of tumors malignancy without any presented statistical data. If you can give relevant statistical data, it will be more credible.

We believe that this result has not been well interpreted or maybe we not understand this comment well.

The results of the study indicate that during the pandemic, the probability that a sample that reaches the laboratory was malignant was higher than in pre-COVID era, and this result is statistically significant. This does not indicate that “COVID-19 pandemic has increased the probability of tumors malignancy” but rather that it is a result of the laboratory's healthcare activity.

 3. Although the authors may make lots of efforts in counting the tumor samples, they didn’t do enough work except for presenting statistical data they obtained and analyzing the impact of the COVID-19 pandemic on cancer diagnosis. It is obviously that the delayed diagnosis may increase the probability of tumors malignancy, you should give more points by the further analysis of samples and explain the reasons. In addition, the impact of COVID-19 on cancer patients should be discussed.

Unfortunately, we do not have data to answer the points made by the reviewer or to analyze the impact of COVID on cancer patients.As indicated in the methodology and limitations of the study, no information is available on patients and, therefore, we cannot distinguish those who had COVID and cancer or only cancer.This work uses totally anonymized data resulting from laboratories activity records and without other clinical information than the SNOMED code for each sample/patient.

On the other hand, the data presented on COVID-19 are non-individualized population data that correspond to the geographical area in which the participating hospitals are located. These data only serve as a reference for hospitalizations due to COVID-19 during the study period.

4. The impact of COVID-19 on patients goes far beyond delayed diagnosis, the SARS-CoV-2 infection may also affect the proliferation of tumor cells by the activation of further immune reactions or the further infection may make patients’ condition worse, these are issues of more concerns. You should take advantage of the obtained resources and get more conclusions.

As we have indicated in the previous answer, the data available do not allow us to analyze the impact of SARS_COV-2 infection in the way that the reviewer comments.

Reviewer 4 Report

Dear authors, the manuscript is well written, clear and concise. Addresses a current topic in Health systems and cancer treatment. However, there are some aspects to be improved:

  • Methodology: the study period must be more detailed and clear (months/ number of months);
  • It is not clear if the characterization of the tumours is comparable, once is one hospital is used SNOMED-CT and in the other was SNOMEDII. The adjustment should be more clear.
  • In table 1, the range proposed above should adjust the comparison.
  • In lines 301-306, the creation of a specific oncologic gynecological unit induces bias on the reported dignosed tumours.
  • In line 324-325, the authors propose new diagnostic methods (other than biopsy) that should be elaborated.

Please rewrite lines 327 to 328: it is not scientific language. 

Author Response

Reviewer 4

Dear authors, the manuscript is well written, clear and concise. Addresses a current topic in Health systems and cancer treatment. However, there are some aspects to be improved:

  • Methodology: the study period must be more detailed and clear (months/ number of months);

Done. The months of the study period has been detailed in the first sentence of Methodology section as follows:

“…The study included all specimens processed in the pathology laboratories of two hospitals in the province of Girona between January  2019 and December 2020:...”

Please note that the months used to compare COVID-19 exposure are already specified in the in the statistical analysis section.

  • It is not clear if the characterization of the tumours is comparable, once is one hospital is used SNOMED-CT and in the other was SNOMEDII. The adjustment should be more clear.

We agree with the reviewer that the consistency of characterization of the tumour between hospitals is not clear. In order to clarify this point we have modified the related paragraphs of methods section as follows:

“…In the two participant laboratories, the pathological study included primary samples from autopsies, cytologies, biopsies and molecular pathology. The results of the pathological study were encoded at Josep Trueta Hospital using SNOMED-CT (Systematized Nomenclature of Medicine - Clinical Terms). SNOMED-CT covers a broad range of health-related topics with comprehensive, scalable, flexible and internationally controlled vocabulary. The Oncology Master Plan has a pathology subset and microglossary, called SNOMED-CT, which has been created using systematically cross-referenced international classifications of diseases, such as the International Classification of Disease Oncology, Third Edition, First revision (ICD-O-3.1), the WHO/IARC classification of tumours series and the TNM Classification of Malignant Tumours 11,12.

A parallel procedure was used to identify the characteristics of the samples (topography, morphology and behaviour when a tumour was present) from Figueres Hospital, the initial encoding of which was performed using SNOMED II that is a previous version of SNOMED-CT.

Using the pathology subset and microglossary from the Oncology Master Plan, the SNOMED-CT and SNOMED II codes were recoded to identify cancer patients and obtain topography and morphology according to ICD-O-3.113. In addition to the topographic and morphological description, when tumours were present it was determined whether these were in situ, infiltrating or metastatic….”

  • In table 1, the range proposed above should adjust the comparison.

As can be seen in the previous answer related to the methodology, the categories used in table 1 are consistent in both participating laboratories. Specifically, the third paragraph of page 3 reads as follows:

“…In the two participant laboratories, the pathological study included primary samples from autopsies, cytologies, biopsies and molecular pathology....”

  • In lines 301-306, the creation of a specific oncologic gynecological unit induces bias on the reported diagnosed tumours.

We agree with the reviewer that the specific oncologic gynecological unit induces bias; in fact, due to this reason we wanted to specify the creation of the unit since it interferes with any possible explanation for the increase in diagnoses in these locations. To make this bias clear, we have rewritten this section, as follows:

Discussion section, page 10:

“...Finally, the interpretation of the increase in cervical and uterine tumours is limited since there was a change in the gynaecological care protocols in Girona due to the creation of a specific oncological gynaecology unit, probably leading to an absence of this cancer diagnosis in county hospitals. This study has other limitations that should be also mentioned....”

  • In line 324-325, the authors propose new diagnostic methods (other than biopsy) that should be elaborated.

Please note that we do not propose new diagnostic methods but we say that the use of other diagnostic methods have been used to diagnose cancer without histological confirmation. To clarify this idea we have rewritten this sentence as follows (Conclusions section):

 “…The results of this study suggest a diagnostic delay in certain types of cancer, such as colorectal or urinary tract tumours, as well as a delay in elective surgery on critical tumours, such as those of the central nervous system or head and neck, which may worsen prognosis. However,  it is possible that the use of specific tumour markers or other diagnostic techniques including X-ray, imaging or ultrasound without waiting histological confirmation, as well as new therapeutic protocols adapted to the new health situation 21,44, may have partially reduced the impact of the pandemic in certain neoplasms ….”

  • Please rewrite lines 327 to 328: it is not scientific language. 

According the reviewer comment, we have redrafted the sentence that now reads as follows:

“...as well as new therapeutic protocols adapted to the new health situation 21,44, may have partially reduced the impact of the pandemic in certain neoplasms…”

Round 2

Reviewer 2 Report

The manuscript has been improved to the point where I recommend acceptance.

Author Response

Thank you. We have reviewed language and style.

Reviewer 4 Report

The authors provide valulable responses to the commnents.

Author Response

(The authors gave the same response as above.)
